# Establishment of Novel High-Grade Serous Ovarian Carcinoma Cell Line OVAR79

**DOI:** 10.3390/ijms252413236

**Published:** 2024-12-10

**Authors:** Polina V. Shnaider, Irina K. Malyants, Olga M. Ivanova, Veronika S. Gordeeva, Ekaterina A. Svirina, Natalya B. Zakharzhevskaya, Olga Y. Shagaleeva, Oksana V. Selezneva, Alexandra N. Bogomazova, Maria M. Lukina, Olga I. Aleshikova, Nataliya A. Babaeva, Andrey V. Slonov, Victoria O. Shender

**Affiliations:** 1Center for Precision Genome Editing and Genetic Technologies for Biomedicine, Lopukhin Federal Research and Clinical Center of Physical-Chemical Medicine of Federal Medical Biological Agency, 119435 Moscow, Russia; 2Lopukhin Federal Research and Clinical Center of Physical-Chemical Medicine of the Federal Medical and Biological Agency, 119435 Moscow, Russia; 3Faculty of Biology, Lomonosov Moscow State University, 119991 Moscow, Russia; 4Institute for Regenerative Medicine, Sechenov University, 119991 Moscow, Russia; 5Shemyakin-Ovchinnikov Institute of Bioorganic Chemistry of the Russian Academy of Sciences, 117997 Moscow, Russia; 6National Medical Scientific Centre of Obstetrics, Gynaecology and Perinatal Medicine Named After V.I. Kulakov, 117198 Moscow, Russia

**Keywords:** high-grade serous ovarian cancer, ovarian cancer, cell line

## Abstract

High-grade serous ovarian carcinoma (HGSOC) remains the most common and deadly form of ovarian cancer. However, available cell lines usually fail to appropriately represent its complex molecular and histological features. To overcome this drawback, we established OVAR79, a new cell line derived from the ascitic fluid of a patient with a diagnosis of HGSOC, which adds a unique set of properties to the study of ovarian cancer. In contrast to the common models, OVAR79 expresses *TP53* without the common hotspot mutations and harbors the rare combination of mutations in both PIK3CA and PTEN genes, together with high-grade chromosomal instability with multiple gains and losses. These features, together with the high proliferation rate, ease of cultivation, and exceptional transfection efficiency of OVAR79, make it a readily available and versatile tool for various studies in the laboratory. We extensively characterized its growth, migration, and sensitivity to platinum- and taxane-based treatments in comparison with the commonly used SKOV3 and OVCAR3 ovarian cell lines. In summary, OVAR79 is an excellent addition for basic and translational ovarian cancer research and offers new insights into the biology of HGSOC.

## 1. Introduction

Ovarian cancer is one of the most lethal gynecological malignancies; it ranks third in incidence among gynecological cancers but first in mortality [1,2]. Lack of robust screening tests for early detection not only reduces the effectiveness of treatment, but also significantly increases its cost. This is one of the reasons why ovarian cancer ranks as the most expensive malignancy to treat among all types of cancer. In the United States, the economic toll of treatment per patient is substantial, with expenses ranging from approximately $80,000 in the first year of care to $100,000 in the terminal stages of the disease [3].

Epithelial ovarian cancer (EOC), which constitutes the majority of ovarian cancer cases, is a very heterogeneous disease characterized by multiple histological subtypes [4]. As a result of early symptoms often being absent, about 75% of EOC cases are diagnosed when the disease has reached an advanced stage, thereby significantly reducing the chance of effective treatment [5]. The overall five-year survival rate following diagnosis is about 46% [6].

EOC has further sub-classifications into subtypes that are based on differences in morphology, etiology, pathogenesis, and molecular biology. High-grade serous ovarian cancer (HGSOC) is the most frequent subtype, accounting for 68–75% of all epithelial ovarian cancer (EOC) cases [2]. This subtype has been characterized by the following characteristics: frequent mutations in key genes, such as *TP53*, *NF1*, *BRCA1*, *BRCA2*, *RB1*, *CDK12*, *FAT3*, and *CSMD3*, coupled with chromosomal copy number variations and genomic duplications [7,8]. The standard treatment approach to HGSOC includes primary cytoreductive surgery followed by platinum-based chemotherapy. However, this EOC subtype is often associated with high chemoresistance [6,9]. This highlights the need to create good preclinical models to study the disease and test new therapeutic strategies. Well-characterized cell lines remain essential for the in vitro study of HGSOC.

However, a major challenge in this field is the issue of cross-contamination and misidentification of cell lines, and this undermines the accuracy of scientific research. For example, it was found that 8 out of 51 ovarian cancer cell lines had been misidentified and were derived from other malignancies such as breast, cervical, or teratocarcinoma cells [10]. Long-term cultivation and immortalization can also introduce new mutations and the selection of particular cell clones, which may result in phenotypic drift away from the characteristics of the original tumor. For example, HeLa (established in 1951 [11]), MCF-7 (established in 1973 [12]), and HEK293 (established in 1973 [13]) have all been reported to show significant variations between strains from different laboratories [14].

These challenges demand new, well-characterized cell lines that maintain the histological and molecular characteristics of HGSOC. A suitable cell line should have high proliferative capacity, be widely available, inexpensive to culture, and amenable to common gene editing techniques.

For comparative purposes, we selected two well-established ovarian cancer cell lines: SKOV3 and OVCAR3. SKOV3 has been an extensively used model for HGSOC, although recent evidence shows that this cell line represents the phenotype of ovarian clear-cell adenocarcinoma [15]. SKOV3 is versatile, as it grows fast, is tolerant to environmental stresses, and its cultivation does not require especial growth conditions or expensive supplements. Furthermore, this cell line can be used as a parental cell line for stem-like cancer cell isolation due to its high levels of stemness markers and chemoresistance [16,17]. On the other hand, OVCAR3 is a more accurate model of HGSOC, but its slower proliferation rate and increased sensitivity to freezing and thawing pose challenges for experimental work.

Here, we report a new cell line, OVAR79, derived from the ascitic fluid of a patient with HGSOC. OVAR79 retains many of the key features of the disease—genetic and phenotypic markers of HGSOC—and offers the advantages of rapid proliferation, stability, and can be maintained on a relatively simple medium without any supplements. This cell line holds great potential as a robust and reliable model for ovarian cancer research.

## 2. Results

### 2.1. Patient Medical History

The OVAR79 cell line was established from the ascitic fluid of a 47-year-old Caucasian woman diagnosed with high-grade serous ovarian carcinoma (HGSOC). Ascitic fluid was collected before therapy by puncture through the posterior vaginal fornix. The diagnosis was confirmed through a comprehensive workup, including MRI, histopathological, and immunohistochemical analyses. Immunocytochemical studies of the ascitic fluid cells revealed a high expression of MOC31, PAX8, CA125, WT1, and p53, consistent with the HGSOC diagnosis. At the time of sample collection, the patient’s disease was classified as stage IV, meaning the cancer was highly advanced.

### 2.2. The Basic Characteristics of the Cell Line

By the 10th passage, the OVAR79 cell line had stabilized and exhibited a homogeneous population with no detectable non-tumor cells. To confirm the authenticity and purity, we conducted short tandem repeat (STR) profiling of OVAR79 cells at the 20th passage and showed that this line is unique and uncontaminated (Figure 1A). The cell line underwent spontaneous immortalization without external intervention, which may have helped preserve key genetic features of the original tumor. Routine mycoplasma testing confirmed the absence of contamination throughout cultivation.

Morphologically, OVAR79 cells are medium-sized and adherent, characterized by the formation of rounded colonies prior to reaching confluency (Figure 1B). The cells have centrally located, large nuclei, which are usually observed in high-grade serous ovarian carcinoma cells. OVAR79 cells prefer to grow in close contact within round-shaped islets and it is easy to detach them from cultural flasks with 0.05% trypsin within a few minutes, but they are stable to mild physical disturbance.

Using the xCELLigence system, we observed that the OVAR79 cells demonstrate a higher biomass accumulation rate compared to the widely used ovarian cancer cell lines SKOV3 and OVCAR3 (Figure 1C). Furthermore, the wound healing assay (Figure 1D,E) revealed that OVAR79 cells exhibit higher proliferation and migration rates compared to the aggressive SKOV3 cell line and OVCAR3.

We compared the transfection efficiency of the SKOV3, OVCAR3, and OVAR79 cell lines. Cells were transfected with the plasmid encoding the spliceosomal protein SRSF2 fused with a red fluorescent protein (pTagRFP-C-SRSF2) using Lipofectamine. We showed that OVAR79 exhibited comparable transfection efficiency to SKOV3 and OVCAR3. Additionally, OVAR79 cells demonstrated higher viability and proliferative activity after transfection compared to SKOV3 and OVCAR3 (Figure 1F).

### 2.3. Mutation Profile

One of the main genetic features of HGSOC is mutation in the *TP53* gene and pronounced genome instability. Additionally, there is a list of genes that are commonly mutated in HGSOC, although the frequency of these mutations is considerably lower. It is believed that the mutational load of HGSOC is primarily associated with genome instability and is its secondary feature [18]. In our study, we analyzed hotspot mutations in genes frequently mutated in HGSOC in particular and ovarian cancer in general, including *TP53*, *KRAS*, *BRAF*, *PIK3C*, *PTEN*, *CTNNB*, *NF1*, and *RB1*.

Sanger sequencing of DNA isolated from the OVAR79 cell line identified mutations in the *PIK3CA* and *PTEN* genes. The *PIK3CA* gene encodes a p110α subunit of the enzyme phosphatidylinositol 3-kinase (PI3K), which plays a critical role in the enzyme’s function. PI3K signaling regulates many processes that are essential for cancer cell survival and proliferation [19]. Specifically, mutations were detected in exons 1 (Leu113Pro) and 9 (Ser553Thr) of the *PIK3CA* gene, affecting the p85 binding site and helical domain, respectively (Figure 2A,B) [20]. Meanwhile, PTEN functions as a suppressor of the PI3K signaling pathway and is frequently mutated in some types of cancer [21]. We found intronic mutations near exons 2 (rs1903858) and 4 (rs1426397261) of the *PTEN* gene, within its phosphatase domain (Figure 2C,D) [22]. Upregulation of the PI3K pathway is a common feature for EOC and leads to chemoresistance acquisition and cell cycle progression. Inhibition of this pathway induces genome instability and mitotic catastrophe by reducing the activity of Aurora kinase B [16].

Mutations in this pathway are frequently associated with aberrant cell signaling and have been implicated in tumor growth and survival, suggesting that OVAR79 may serve as a useful model for studying PI3K/AKT pathway dysregulation in ovarian cancer [23,24].

The *TP53* gene encodes the p53 protein and is frequently mutated in many types of cancer due to its critical role as a tumor suppressor. This gene plays a central role in maintaining genomic stability by regulating various cellular processes, including apoptosis, the cell cycle, and DNA repair [25]. Although *TP53* mutations are present in approximately 96% of high-grade serous ovarian cancer cases, no mutations were detected in the hotspot regions of exons 3–8 within the OVAR79 cells [7]. Immunocytochemical analysis confirmed wild-type p53 expression in the original clinical ascites sample. This rarity makes the OVAR79 line particularly valuable for exploring the functions of p53 and its downstream regulatory pathways, including its roles in DNA repair, apoptosis, and cell cycle control.

No mutations were detected in the *KRAS* (exons 1 and 2), *BRAF* (exon 15), *CTNNB1* (exon 2), *NF1* (exons 3, 5, 37, and 39), or *RB1* (exons 17 and 20) genes. The absence of mutations in these genes further underscores the potential utility of the OVAR79 cell line as a unique tool for research on wild-type p53 HGSOC.

### 2.4. DNA Copy Number Alterations

HGSOC cannot be characterized by specific somatic mutations due to high variability among patients. However, HGSOC is associated with numerous gains and losses of chromosomal material [7]. To investigate copy number alterations in the OVAR79 cell line, we performed karyotypic analysis.

SNP array genotyping analysis of the OVAR79 cell line revealed numerous chromosomal rearrangements. We observed frequent large-scale copy number losses common to this ovarian cancer subtype, with deletions in chromosome 4p occurring in 65–67% of cases, 8p in 74%, 12q in 35%, and 13q in 64%. Additionally, we noted the loss of the X chromosome and a relatively high level of copy-neutral loss of heterozygosity (cnLOH), affecting approximately 41% of the genome (Figure 3).

Most of the identified chromosomal imbalances overlap with previously described regions of genomic instability in HGSOC. For comparison, we used data on both focal gains and losses as well as on gene copy number alterations (Appendix A) [7,26,27]. Among the identified copy number losses, we observed several key loci, including the tumor suppressor gene *BRCA2* and its partner *PALB2*, which both are essential for DNA repair; the anti-oncogene *RB1*, which regulates cell proliferation; *PTCH1*, encoding a crucial signaling receptor in the sonic hedgehog pathway; and *ARID1A*, which influences DNA repair through chromatin organization [28,29,30,31]. Additionally, we found amplification of 13 genes that represent known therapeutic targets, including *PAX8*, *MYC*, *AKT3*, *ID4*, and *MALAT1*, in the OVAR79 cell line [32,33,34,35].

### 2.5. Marker Expression and Phenotypic Analysis of OVAR79 Cells

The OVAR79 cells displayed an epithelial phenotype based on the expression levels of the epithelial markers CDH1 (CD324, E-cadherin) and ZO1. In contrast, RT-qPCR showed very low expression of mesenchymal markers such as CDH2 (CD325, N-cadherin) and VIM (Vimentin). At the same time, we noticed the upregulation of SNAIL, one of the major transcription factors related to epithelial–mesenchymal transition (EMT), which may suggest that this cell line is in a transitional state and could be prone to partial EMT changes under specific conditions, thus expressing an increase in cellular aggressiveness [36]. Compared to the SKOV3 and OVCAR3 cell lines, OVAR79 cells show an intermediate level of epithelial characteristics. The OVAR79 line shows significant expression of NANOG, a key pluripotency factor [37]. However, the relative expression levels of other stemness markers, such as ALDH1A1, POU5F1 (Oct-4), and SOX2, are very low [38,39]. High CD44 expression in OVAR79 cells may correlate with increased proliferation (Figure 1C) and therapy resistance (Figure 4A) [36,40].

Flow cytometry analysis further supports the epithelial nature of the OVAR79 cell line, demonstrating a higher expression of epithelial markers such as CD324 (E-cadherin), CA125, and EpCAM, compared to the mesenchymal markers CD325 (N-cadherin) and Vimentin (Figure 4B). The surface marker CD44 is also prominently expressed, aligning well with the RT-qPCR data (Figure 4A). Additionally, high levels of cytokeratins 4, 5, 6, 8, 10, 13, and 18 were observed through PanCK staining, further confirming the epithelial phenotype (Figure 4B).

### 2.6. In Vitro Chemosensitivity

A cytotoxicity assay was performed for the estimation of relative sensitivity of the cell lines OVAR79, SKOV3, and OVCAR3 to known pharmacological agents utilized for ovarian cancer treatment. As shown in Figure 4C, the OVAR79 cell line was more sensitive to low concentrations of cisplatin, with the same value of half-maximal inhibitory concentration (IC50) as SKOV3 and OVCAR3 cells. On the other hand, the cell line OVAR79 showed higher resistance against carboplatin and paclitaxel.

### 2.7. Comparison of OVAR79 to Other Ovarian Cancer Cell Lines

The majority of ovarian cancer research utilizes cell lines such as SKOV3, OVCAR3, A2780, and CaOV3 [41]. In our study, we compared the features of the OVAR79 cell line with SKOV3 and OVCAR3 cell lines through direct experiments. To address the limitation of using a restricted number of cell lines, we referred to published data to include comparative characteristics of A2780 and CaOV3 cell lines. We expanded the comparison by incorporating key aspects of cell line characteristics, including patient diagnosis, origin of the cell line, mutational burden and chromosomal instability, proliferation and migration rates, and epithelial–mesenchymal transition status (Table 1). Despite the limited set of descriptive characteristics, all these cell lines are completely different and do not duplicate each other. At the same time, it is well known that some of the most commonly used ovarian cancer cell lines do not represent HGSOC, despite their frequent use as HGSOC models in numerous studies. In summary, this study highlights the importance of accurately characterizing new cell lines and carefully selecting a disease model for each research task based on the specific characteristics of the cell line.

## 3. Discussion

Despite the frequent occurrence of the disease, cell lines representing HGSOC are often under-characterized and sometimes misidentified. The main genetic characteristics of HGSOC include *TP53* mutations, homologous recombination deficiency, and a high level of genomic alterations [41,47]. HGSOC is also characterized by the expression of epithelial phenotype markers such as MUC16 (CA125), MSLN, PAX8, and KRT7 [53].

HGSOC is a markedly heterogeneous disease with significant diversity. Individual subtypes present exclusive features, but these may not always correspond to strict boundaries. For example, the presence or absence of a *TP53* mutation alone is not a strict criterion for determining the subtype of a tumor. A biobank of ovarian cancer primary cell cultures, created by Nelson L. et al., revealed that genomic instability (the main characteristic of HGSOC) can be maintained even in the presence of wild-type *TP53*. Furthermore, it became evident that, despite having the same diagnosis, each tumor represents a unique combination of genetic alterations and mutations across a wide spectrum of genes [59]. It is worth noting that, despite their high genetic similarity, cancer cells from ascites differ from the original ovarian tumor cells. This difference arises because ascites have distinct physical properties and a unique cellular and molecular context compared to the solid tumor, creating selective conditions for clones capable of surviving without adhesion. Moreover, cancer cells in ascites can form spheroids, spherical cell conglomerates, that exhibit higher metastatic potential, increased expression of genes related to chemoresistance, and upregulation of the oxidative phosphorylation pathway [60,61,62]. Another important point is that cancer cells derived from ascites tend to display a more mesenchymal phenotype, which contributes to their higher metastatic potential, compared to the original tumor cells [63]. Therefore, the origin of a cancer cell line should be carefully taken into account.

We established the OVAR79 ovarian cancer cell line originating from ascites of a patient who was clinically diagnosed with high-grade serous ovarian carcinoma (HGSOC). Although *TP53* mutations are predominant in HGSOC cases, we found no mutations in *TP53* hotspots in OVAR79 cells. Clinical tests have confirmed the presence of p53 protein in the tumor tissue. However, mutations were found in the *PTEN* and *PIK3CA* genes. We believe that our cell line represents the HGSOC subtype, which is known for widespread genomic alterations marked by prominent copy number changes, and we showed that 42% of the genome is affected in OVAR79.

Our results show that OVAR79 cells exhibit an epithelial phenotype, including some features related to stemness as well as a more aggressive phenotype. Moreover, OVAR79 cells are more sensitive to cisplatin but more resistant to carboplatin compared with SKOV3 and OVCAR3 cells, as well as more resistant to paclitaxel compared to OVCAR3. However, for greater reliability, these results should be validated in other independent laboratories.

A major feature of OVAR79 is its simple handling. The cells have high proliferative activity, rapidly accumulate biomass, have a high transfection efficiency, and do not require expensive culture media or supplements. They are resistant to common environmental stressors, adhere well to the culture plastic, and can easily be detached with 0.05% trypsin without being dislodged by gentle mechanical agitation, and they maintain a stable phenotype over long cultivation periods.

The OVAR79 cell line is a new and convenient model for the study of ovarian cancer and unique phenotypes of HGSOC, especially in those cases with a lack of *TP53* mutations but with a large number of genomic alterations.

## 4. Materials and Methods

### 4.1. Patient and Sample Data

Ascitic fluid was removed from a 47-year-old patient diagnosed with stage IV high-grade serous ovarian cancer (HGSOC) prior to chemotherapy through the posterior vaginal fornix at the Russian Scientific Center of Roentgenoradiology under the Ministry of Health of Russia. Ethical approval was obtained from the Ethics Committee of the National Medical Research Center for Obstetrics, Gynecology, and Perinatology named after Academician V.I. Kulakov of the Ministry of Healthcare of the Russian Federation (protocol no. 10 of 5 December 2019). The patient provided written informed consent for participation.

### 4.2. Cell Line Establishment and Culture Conditions

The human ovarian cancer cell line SKOV3 (ATCC, HTB-77) was grown in DMEM medium (Paneco, Moscow, Russia) supplemented with 10% FBS (Gibco/Thermo Fisher Scientific, Waltham, MA, USA), 2 mM l-glutamine (Paneco, Moscow, Russia), and 1% penicillin/streptomycin (Paneco, Moscow, Russia). The human ovarian cancer cell line OVCAR3 (ATCC, HTB-161) was grown in RPMI medium supplemented with 10% FBS, 2 mM l-glutamine, and 1% penicillin/streptomycin.

Ascitic fluid was collected in a sterile 50 mL tube and processed within an hour to sediment and isolate tumor cells. For this, the ascites was centrifuged at 200× *g* for 15 min at 4 °C. Cells from the ascites were resuspended in 5 mL of RPMI nutrient medium (Gibco/Thermo Fisher Scientific, Waltham, MA, USA) supplemented with 10% FBS, 2 mM glutamine, and 1% penicillin/streptomycin, with the addition of 1% non-essential amino acids (Paneco, Moscow, Russia). The cell line was assigned the identification number OVAR79, and currently, the cells have reached more than 100 passages. All studies on OVAR79 cells were conducted on a homogeneous cell line that had reached passage 30. The cell lines were regularly tested for mycoplasma contamination. All cell lines were incubated at 37 °C in a humidified atmosphere containing 5% CO_2_.

### 4.3. Short Tandem Repeat (STR) Analysis

Short tandem repeat (STR) analysis were performed on the OVAR79 cell line for 19 markers (D3S1358, D5S818, D7S820, D8S1179, D13S317, D16S539, D18S51, D21S11, CSF1PO, FGA, TH01, TPOX, VWA, D1S1656, D2S441, D10S1248, D12S391, D22S1045, SE33) and amelogenin loci using the COrDIS Plus kit according to the manufacturer’s instructions (Gordiz Ltd., Moscow, Russia). Analysis of DNA fragments was performed with an ABI3130 Genetick Analyzer (Applied Biosystems/Thermo Fisher Scientific, Waltham, MA, USA).

### 4.4. Cell Proliferation Measurement

Proliferation rate was measured using a xCELLigence RTCA Dual Plate (RTCA-DP) instrument according to the manufacturer’s instructions (Agilent Technologies, Santa Clara, CA, USA). Cells were seeded on an E-plate 16 (2000 cells/well for OVAR79, 2500 cells/well for OVCAR3, 2000 cells/well for SKOV3) and placed in an RTCA DP Analyzer maintained at 37 °C and 5% CO_2_ for 185 h. The electrical impedance values, represented as the cell index, were then recorded at 15 min intervals over a 7-day period.

### 4.5. Wound Healing Assay

Cells were seeded into a 4-well insert dish (Ibidi, Gräfelfing, Germany), and upon reaching confluence, the insert was removed to create a wound in the center of the monolayer. Cells were washed with PBS to remove cell debris. Images of the cells were taken immediately after insert removal, and then after 8 and 24 h, using an Olympus IX53F (Olympus, Tokyo, Japan) fluorescence microscope.

### 4.6. Single Nucleotide Polymorphism (SNP) Array Analysis

Genomic DNA was processed according to the Infinium HD assay protocol (Illumina, San Diego, CA, USA) and hybridized on a CytoSNP-12v2 Beadchip array (Illumina, San Diego, CA, USA). Screening for genomic abnormalities was performed using BlueFuse Multi software v 4.4 with default settings. Two datasets were used to assess consistency: copy number profiles of 489 HGS-OvCa TCGA samples and copy number alterations from targeted and whole-exome sequencing of 45 patients with HGSOC [7,27].

### 4.7. Mutation Identification

The cell line samples were washed three times in PBS buffer, collected, and stored at −20 °C until genetic study. DNA was isolated with a QIAamp DNA mini kit (Qiagen, Hilden, Germany) and amplified using targeted primers. The primer sequences were partially obtained from an article by Ying-Cheng Chiang [64]. Other primers were designed with Primer-BLAST (provided in the public domain by the National Center for Biotechnology Information, Bethesda, MD, USA). The primers for the relevant exons of the *KRAS*, *BRAF*, *TP53*, *PIK3CA*, *PTEN*, *CTNNB*, *NF1*, and *RB1* genes are listed in Table 2.

The applied PCR conditions were as follows: reaction volume, 25 μL; DNA input, 15 ng; Encyclo Plus PCR kit (Evrogen, Moscow, Russia); and each primer at a final concentration of 5 pM. Sanger sequencing was carried out according to the manufacturer’s protocol with the BigDye Terminator v3.1 Kit (Applied Biosystems/Thermo Fisher Scientific, Waltham, MA, USA) on an ABI Prism 3730XL Genetic Analyzer (Applied Biosystems/Thermo Fisher Scientific, Waltham, MA, USA). ABI sequencing files were analyzed in Sequence Scanner Software (2.0).

### 4.8. Reverse Transcription Followed by Semi-Quantitative PCR

Total RNA was extracted using RLT buffer with 40 mM DTT, followed by processing and purification according to the RNeasy Mini Kit protocol (Qiagen, Hilden, Germany). RNA concentration was measured using a Qubit Fluorometer (Invitrogen/Thermo Fisher Scientific, Waltham, MA, USA). cDNA was synthesized using iScript reverse transcription supermix (Bio-Rad Laboratories, Hercules, CA, USA) according to the manufacturer’s instructions. RT-qPCR reactions were carried out using qPCRmix-HS SYBR (Evrogen, Moscow, Russia) on the CFX96 Touch Real-Time PCR Detection System (Bio-Rad Laboratories, Hercules, CA, USA). All samples were tested in triplicate, and GAPDH was used as a reference gene for data normalization. PCR products and primer specificity were verified via electrophoresis on a 1.5% agarose gel. Primers used for PCR are listed in Table 3.

### 4.9. FACS Analysis

Cells were detached using 0.05% trypsin, washed three times with PBS, and fixed with 4% paraformaldehyde (Sigma-Aldrich, St. Louis, MO, USA). The cells were then placed in a serum-free protein block solution ((Dako/Agilent Technologies, San Diego, CA, USA) and incubated with primary antibodies—anti-CD324 (E-cadherin; Sony, Tokyo, Japan, #1336505), anti-EpCAM (Abcam, Cambridge, UK, ab223582), anti-CD44 (BD Biosciences, Franklin Lakes, NJ, USA, #2294010), anti-PanCK (Eagle BioSciences, Amherst, NH, USA, 10-310015-01), anti-CA125 (R&D Systems, Minneapolis, MN, USA, mab56092), anti-Vimentin (Thermo Fisher Scientific, Waltham, MA, USA, RM-9120-S0), or anti-CD325 (N-cadherin; Sony, Tokyo, Japan, #2354010)—for 1 h, followed by a 1 h incubation with Alexa Fluor 488- (Invitrogen/Thermo Fisher Scientific, Waltham, MA, USA, #A-11008, #A-11001), Alexa Fluor 555- (Invitrogen/Thermo Fisher Scientific, Waltham, MA, USA, #A-21434), or Alexa Fluor 647-conjugated secondary antibodies (Invitrogen/Thermo Fisher Scientific, Waltham, MA, USA, #A-21235). After staining, samples were analyzed using a NovoCyte Flow Cytometer (ACEA Biosciences/Agilent Technologies, San Diego, CA, USA), and data were processed with NovoExpress Software (1.6.0).

### 4.10. MTT Assay

Cells were seeded onto a 96-well plate at a density of 3500 cells/well for OVAR79 and OVCAR3, and 4000 cells/well for SKOV3, and incubated overnight at 37 °C and 5% CO_2_. Different concentrations of carboplatin (Thermo Fisher Scientific, Waltham, MA, USA, AAJ6043303), cisplatin (Teva Pharmaceutical Industries, Tel Aviv, Israel, N011590/02), or paclitaxel (Cell Signaling Technology, Danvers, MA, USA, 9807S) were added to the cells for 48 h in triplicate. Cell viability was determined using MTT reagent (Sigma-Aldrich, St. Louis, MO, USA). Optical density at 565 nm was measured with an iMark Microplate Reader (Bio-Rad Laboratories, Hercules, CA, USA). Cell survival percentage was calculated by comparing the absorbance of drug-treated cells to that of untreated control cells.

### 4.11. Transient Transfection

To compare the efficiency of transfection, OVAR79, OVCAR3, and SKOV3 cells, seeded on 96-well chamber slides, were transfected with plasmid pTagRFP-C-SRSF2 using Lipofectamine 3000 (Invitrogen/Thermo Fisher Scientific, Waltham, MA, USA, #L3000015) according to the manufacturer’s protocol. After 48 h, the transfected cells were visualized using CELENA X (Logos Biosystems, Anyang, Republic of Korea). Plasmid pTagRFP-C-SRSF2 was kindly gifted by Dr. Marat S. Pavlyukov.

## Figures and Tables

**Figure 1 ijms-25-13236-f001:**
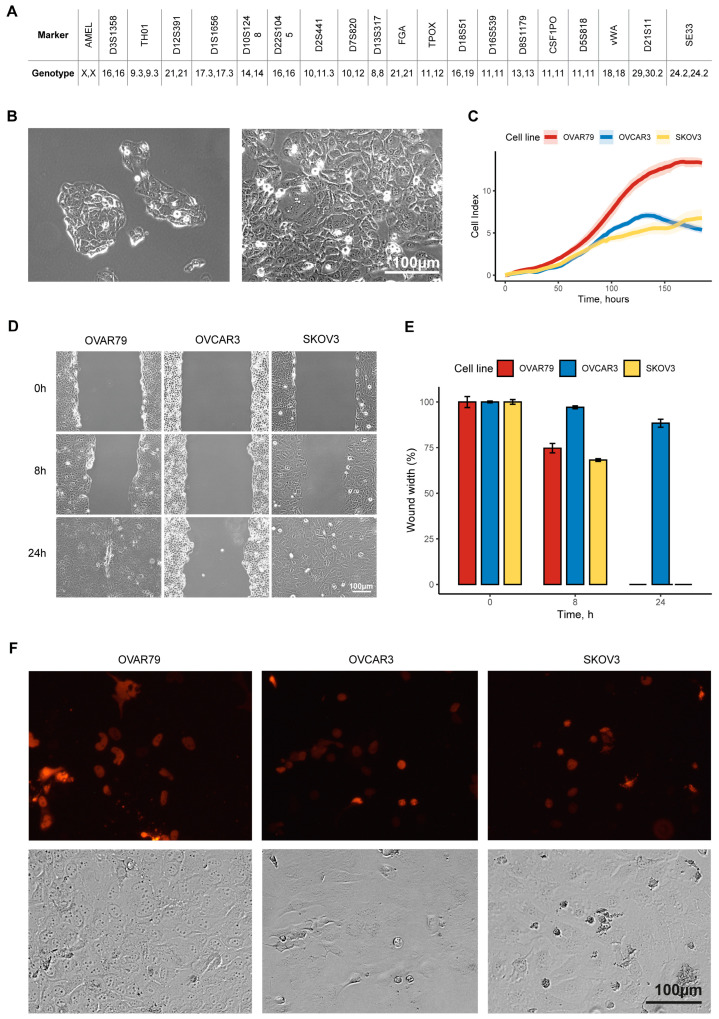
Morphological, growth, and migratory characteristics of the OVAR79 cell line in comparison to SKOV3 and OVCAR3 cell lines. (**A**) Short tandem repeat (STR) profiling of the OVAR79 cell line. (**B**) Representative phase-contrast images of OVAR79 cells at low (left image) and high (right image) confluency. (**C**) Growth curves of OVAR79, SKOV3, and OVCAR3 cell lines. The proliferation rates were measured over 185 h with regular time-point assessments. The data represent the mean ± SD from 3 biologically independent replicates. (**D**,**E**) Wound healing assay of OVAR79, SKOV3, and OVCAR3 cell lines. The width of the wound area was measured immediately after scratching (0 h), with wound closure quantified after 8 and 24 h. The bar graph (**D**) illustrates the average wound closure at each time-point, represented as mean ± SD (*n* = 3 biologically independent replicates). (**F**) Fluorescence and phase-contrast images of OVAR79, OVCAR3 and SKOV3 cell lines expressing SRSF2-RFP protein (red).

**Figure 2 ijms-25-13236-f002:**
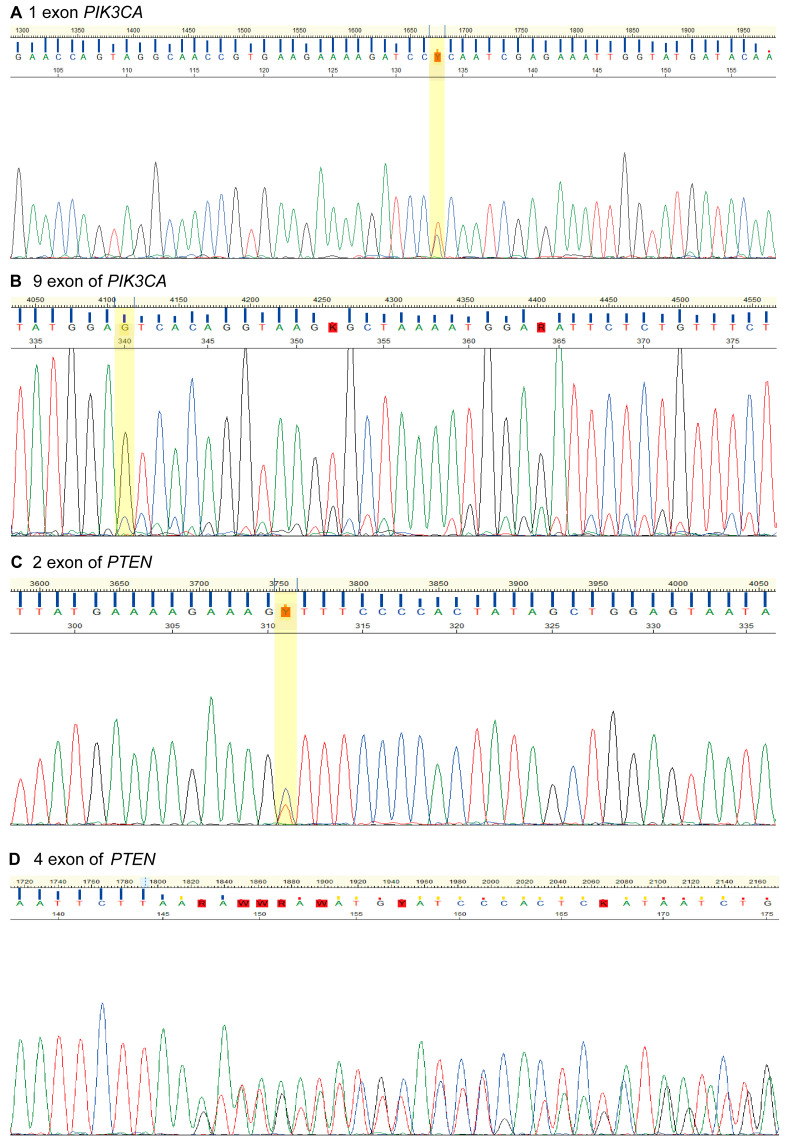
Sanger sequencing electropherograms of the OVAR79 cell line showing mutations in the *PIK3CA* and *PTEN* genes. (**A**) A single-nucleotide substitution in exon 1 (c.338T>C) leading to a codon change (CTC>CCC) and an amino acid substitution (Leu113Pro). (**B**) An indel mutation in exon 9 with a GT deletion and C insertion at position 1658–1659 leading to a codon change (AGT>ACC) and a frameshift. (**C**) A single-nucleotide variant in intron 2, rs1903858 at chr10:87893929. (**D**) An indel mutation around exon 4, rs1426397261, due to a duplication of ATACATATT to ATACATATTATACATATT, located at chr10:87931188-87931196.

**Figure 3 ijms-25-13236-f003:**
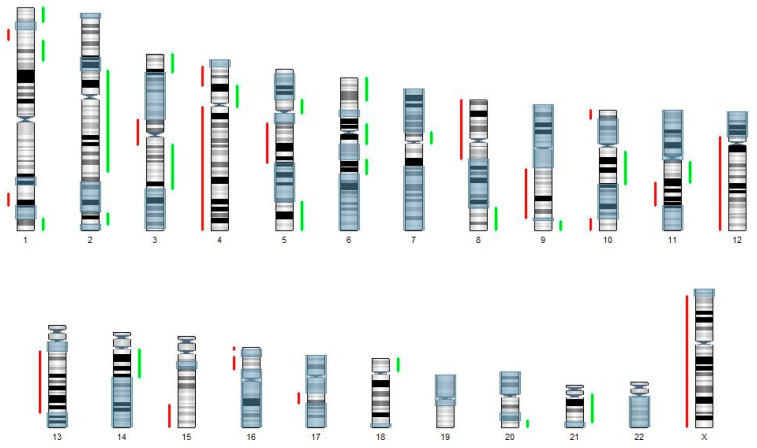
Karyotype of the OVAR79 cell line. Regions of gain are indicated in green, regions of loss in red, and copy-neutral losses of heterozygosity (cnLOH) are shown in light blue.

**Figure 4 ijms-25-13236-f004:**
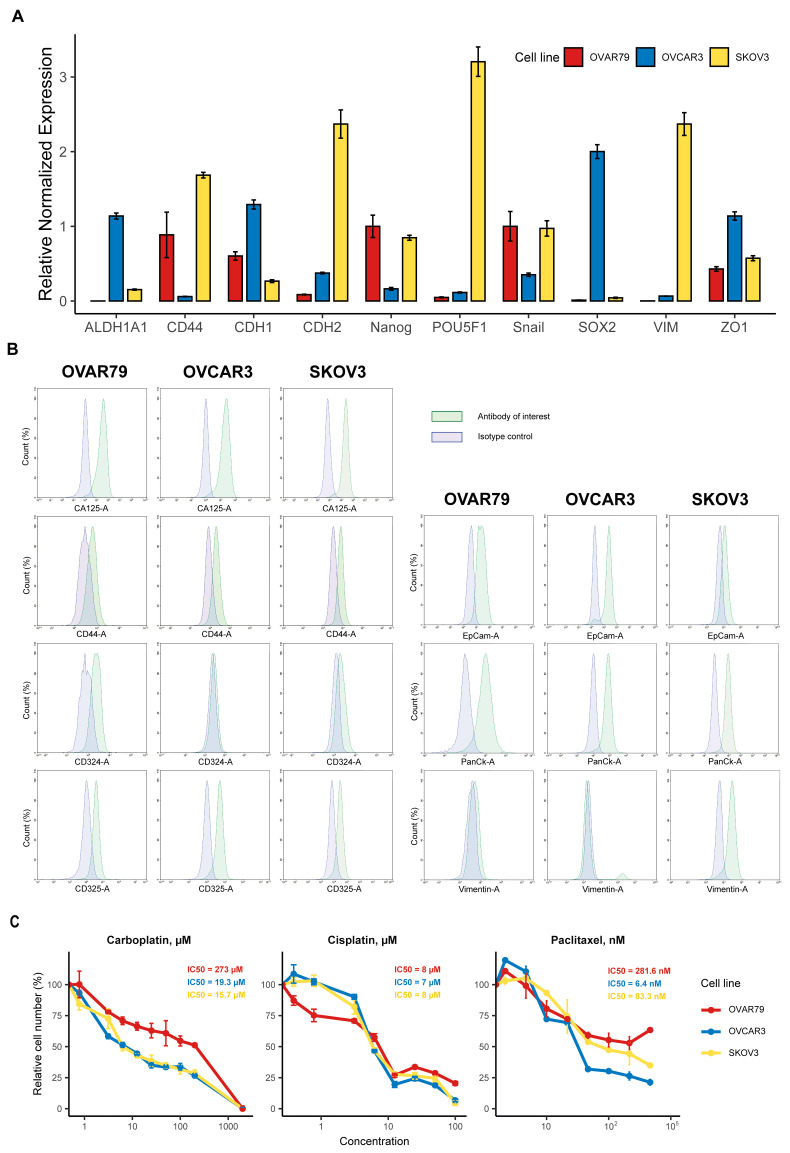
(**A**) RT-qPCR analysis of mRNA expression levels of epithelial, mesenchymal, and stemness markers in OVAR79, SKOV3, and OVCAR3 cell lines. Bars represent the relative expression levels of each marker normalized to GAPDH (*n* = 3 biologically independent experiments). Data represent the mean values ± SEM. (**B**) Flow cytometry analysis of ovarian cancer markers in OVAR79, SKOV3, and OVCAR3 cell lines. The green color represents the antibody of interest, while the blue color corresponds to the respective isotype control for each antibody. Graphs display marker intensity on the x-axis and the percentage of the cell population on the y-axis (%). (**C**) Dose–response curves obtained by MTT assay of OVAR79, OVCAR3, and SKOV3 cells that were treated with different concentrations of cisplatin, carboplatin, or paclitaxel for 48 h. The data represent the mean values ± SD (*n* = 3 biologically independent replicates). IC50 values were determined by fitting a normalized dose–response model to the data using nonlinear regression in GraphPad Prism 8.0 software.

**Table 1 ijms-25-13236-t001:** Comparison of OVAR79 cell line characteristics with the most commonly used ovarian cancer cell lines.

Feature	OVAR79	SKOV3	OVCAR3	CaOV3	A2780
Origin	Ascitic fluid	Ascitic fluid [42]	Ascitic fluid [43]	Tumor [44]	Tumor [45]
Diagnosis	HGSOC	Clear-cell carcinoma [15]	HGSOC [46]	Possibly HGSOC [41]	Endometrioid ovarian carcinoma [47]
*TP53* mutation	Wild-type	Wild-type [41] or truncating mutation [47]	Mutated [46]	Mutated [41]	Wild-type [41]
*PIK3CA* mutations	Mutated	Mutated [47]	Wild-type [48,49]	Wild-type [47]	Mutated [47]
*PTEN* mutations	Mutated	Wild-type [47]	Wild-type [48,49]	Wild-type [47]	Mutated [47]
Proliferation rate	High	High	Moderate	High [50]	High [51]
Migration ability	High	High	Low	High [50]	Low [52]
Epithelial/Mesenchymal markers	Epithelial with EMT potential	Predominantly mesenchymal	Epithelial [53]	Epithelial [53]	Epithelial [53]
Chemoresistance	Highly resistant to carboplatin, moderate sensitivity to cisplatin	Resistant to paclitaxel, resistant to low doses of cisplatin	Resistant to low doses of cisplatin	Resistant to paclitaxel (comparable to SKOV3) [54]	Sensitive to chemotherapy drugs of different classes [55]
Stemness markers	High NANOG, moderate CD44	Moderate	High CD44	Moderate CD44, Bmi-1, low Dclk-1 [56]	Low [57]
Chromosomal instability	High	Moderate [58]	High [58]	Moderate [58]	Low [58]

**Table 2 ijms-25-13236-t002:** Primer sequences used for mutation identification.

Gene	Exon		Primers
*KRAS*	1	Forward primer sequence 5′>3′	5′-TTGAAACCCAAGGTACATTTCAG-3′
		Reverse primer sequence 5′>3′	5′-TCTTAAGCGTCGATGGAGGAG-3′
*KRAS*	2	Forward primer sequence 5′>3′	5′-TATGCATGGCATTAGCAAAGAC-3′
		Reverse primer sequence 5′>3′	5′-CGTCATCTTTGGAGCAGGAAC-3′
*BRAF*	15	Forward primer sequence 5′>3′	5′-TCATAATGCTTGCTCTGATAGGA-3′
		Reverse primer sequence 5′>3′	5′-GGCCAAAAATTTAATCAGTGGA-3′
*TP53*	3–4	Forward primer sequence 5′>3′	5′-GAGGAATCCCAAAGTTCCAAAC-3′
		Reverse primer sequence 5′>3′	5′-ACGTTCTGGTAAGGACAAGGG-3′
*TP53*	5–6	Forward primer sequence 5′>3′	5′-CAGGAGGTGCTTACGCATGTT-3′
		Reverse primer sequence 5′>3′	5′-AGGAGAAAGCCCCCCTACTG-3′
*TP53*	7	Forward primer sequence 5′>3′	5′-AGAAATCGGTAAGAGGTGGGC-3′
		Reverse primer sequence 5′>3′	5′-CATCCTGGCTAACGGTGAAAC-3′
*TP53*	8	Forward primer sequence 5′>3′	5′-TTGGGCAGTGCTAGGAAAGAG-3′
		Reverse primer sequence 5′>3′	5′-GTTGGGAGTAGATGGAGCCTG-3′
*PIK3CA*	1	Forward primer sequence 5′>3′	5′-CCCCTCCATCAACTTCTTCAA-3′
		Reverse primer sequence 5′>3′	5′-ATTGTATCATACCAATTTCTCGATTG-3′
*PIK3CA*	1	Forward primer sequence 5′>3′	5′-TGCTTTGGGACAACCATACATC-3′
		Reverse primer sequence 5′>3′	5′-CTTGCTTCTTTAAATAGTTCATGCTTT-3′
*PIK3CA*	9	Forward primer sequence 5′>3′	5′-TCAGCAGTTACTATTCTGTGACTGG-3′
		Reverse primer sequence 5′>3′	5′-TGCTGAGATCAGCCAAATTCA-3′
*PIK3CA*	20	Forward primer sequence 5′>3′	5′-GACATTTGAGCAAAGACCTGAAG-3′
		Reverse primer sequence 5′>3′	5′-TGGATTGTGCAATTCCTATGC-3′
*PTEN*	1	Forward primer sequence 5′>3′	5′-TTTCCATCCTGCAGAAGAAGC-3′
		Reverse primer sequence 5′>3′	5′-TCCGTCTAGCCAAACACACC-3′
*PTEN*	2	Forward primer sequence 5′>3′	5′-TCTGTGATGTATAAACCGTGAGTTTC-3′
		Reverse primer sequence 5′>3′	5′-CCCTGAAGTCCATTAGGTACGG-3′
*PTEN*	3	Forward primer sequence 5′>3′	5′-ATTACTACTCTAAACCCATAGAAGG-3′
		Reverse primer sequence 5′>3′	5′-TCAAATATGGGCTAGATGCCA-3′
*PTEN*	4	Forward primer sequence 5′>3′	5′-ATAAAGATTCAGGCAATGTTTGTTAG-3′
		Reverse primer sequence 5′>3′	5′-GACCAACTGCCTCAAATAGTAGG-3′
*PTEN*	5	Forward primer sequence 5′>3′	5′-TGCAACATTTCTAAAGTTACCTACTTG-3′
		Reverse primer sequence 5′>3′	5′-TTTACTTGTCAATTACACCTCAATAAA-3′
*PTEN*	6	Forward primer sequence 5′>3′	5′-AATGGCTACGACCCAGTTACC-3′
		Reverse primer sequence 5′>3′	5′-TTTGGCTTCTTTAGCCCAATG-3′
*PTEN*	7	Forward primer sequence 5′>3′	5′-TGCAGATACAGAATCCATATTTCG-3′
		Reverse primer sequence 5′>3′	5′-AATGTCTCACCAATGCCAGAG-3′
*PTEN*	8	Forward primer sequence 5′>3′	5′-TGCAACAGATAACTCAGATTGCC-3′
		Reverse primer sequence 5′>3′	5′-TGTCAAGCAAGTTCTTCATCAGC-3′
*PTEN*	9	Forward primer sequence 5′>3′	5′-AAGATCATGTTTGTTACAGTGCTTAAA-3′
		Reverse primer sequence 5′>3′	5′-TGACACAATGTCCTATTGCCA-3′
*CTNNB*	2	Forward primer sequence 5′>3′	5′-GCGTGGACAATGGCTACTCAA-3′
		Reverse primer sequence 5′>3′	5′-GGATCTGCATGCCCTCATCTA-3′
*NF1*	3	Forward primer sequence 5′>3′	5′-TGTGTGTTGATTGGTAGCAGA-3′
		Reverse primer sequence 5′>3′	5′-AGACAGATACGTGGCTGAAACA-3′
*NF1*	5	Forward primer sequence 5′>3′	5′-TTCTCCACTTCACCCCGTCA-3′
		Reverse primer sequence 5′>3′	5′-AATACCTGCCCAAGGCTTCC-3′
*NF1*	37	Forward primer sequence 5′>3′	5′-TTCTCCACTTCACCCCGTCA-3′
		Reverse primer sequence 5′>3′	5′-ACCTACCGTAAACTCGGGTC-3′
*NF1*	39	Forward primer sequence 5′>3′	5′-TCTCCAGGCCTGATTCTAGGT-3′
		Reverse primer sequence 5′>3′	5′-AATACCTGCCCAAGGCTTCC-3′
*RB1*	17	Forward primer sequence 5′>3′	5′-CTTTCCCATGGATTCTGAATGTGC-3′
		Reverse primer sequence 5′>3′	5′-AGATGGTTTAGGGTGCTCGAT-3′
*RB1*	20	Forward primer sequence 5′>3′	5′-CTTCCACCAGGGTAGGTCAAAA-3′
		Reverse primer sequence 5′>3′	5′-ATAGATTTTCTTCACCCCGCCC-3′

**Table 3 ijms-25-13236-t003:** Primer sequences used for semi-quantitative reverse-transcription PCR.

Gene	Forward Primer	Reverse Primer
*CD324*	GAACGCATTGCCACATACAC	GAATTCGGGCTTGTTGTCAT
*ZO1*	CAACATACAGTGACGCTTCACA	CACTATTGACGTTTCCCCACTC
*CD325*	TGTTTGACTATGAAGGCAGTGG	TCAGTCATCACCTCCACCAT
*VIM*	ATCCAAGTTTGCTGACCTCTC	CTCAGTGGACTCCTGCTTTG
*SNAI1*	CTTCCAGCAGCCCTACGAC	CGGTGGGGTTGAGGATCT
*SOX2*	TTTGTCGGAGACGGAGAAGC	CCCGCTCGCCATGCTATT
*OCT4*	GGGGGTTCTATTTGGGAAGGTA	ACTGGGCGATGTGGCTGAT
*CD44*	CGTGGAATACACCTGCAAAGC	CGGACACCATGGACAAGTTTT
*ALDH1A1*	CCACTCACTGAATCATGCCA	CTGAGCCAGTCACCTGTGTTC
*NANOG*	AATGGTGTGACGCAGGGATG	GCAGGAGAATTTGGCTGGAAC

## Data Availability

Data are contained within the article or Appendix A.

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
