# Peer review of "Establishment of Novel High-Grade Serous Ovarian Carcinoma Cell Line OVAR79"

_ijms, 2024, doi:10.3390/ijms252413236_

Round 1
Reviewer 1 Report
Comments and Suggestions for Authors
In this article, the authors report a new cell line, OVAR79, derived from the ascitic fluid of a patient with HGSOC, which holds great potential as a robust and reliable model for ovarian cancer research. The manuscript is straightforward, well written, and concise. Definitely deserves to be published and is a valuable contribution to the “International Journal of Molecular Sciences”. However, the following comments need to be addressed, as recommended.
[1] “1. Introduction”, Page 1 of 15, Lines 36-37:
“Ovarian cancer is one of the most lethal gynecological malignancies, it ranks third in incidence among gynecological cancers but first in mortality [1,2].”.
At that point, it should be added that there are still no effective tools for general population screening. This is also reflected economically and cost-effective strategies for early detection and prevention of ovarian cancer have been investigated over the last decade. The cost of treatment per patient with ovarian cancer remains the highest among all cancer types. As an example, the average initial cost in the first year can amount to around USD 80,000, whereas the final year cost may increase to USD 100,000.
Recommended reference: Ghose A, et al. Hereditary Ovarian Cancer: Towards a Cost-Effective Prevention Strategy. Int J Environ Res Public Health. 2022;19(19):12057.
[2] “1. Introduction”, Page 2 of 15, Lines 50-52:
“However, this EOC subtype is often accociated with high chemoresistantance [5,8]. This highlights the need to create good preclinical models to study the disease and test new therapeutic strategies.”.
Within this context, it is important to comment that PI3K pathway is frequently upregulated in EOC and plays an important role in chemoresistance and preservation of genomic stability, as it is implicated in many processes of DNA replication and cell cycle regulation. The inhibition of the PI3K may lead to genomic instability and mitotic catastrophe through a decrease of the activity of the spindle assembly checkpoint protein Aurora kinase B and consequently increase of the occurrence of lagging chromosomes during prometaphase.
Recommended reference: Aliyuda F, et al. Advances in Ovarian Cancer Treatment Beyond PARP Inhibitors. Curr Cancer Drug Targets. 2023;23(6):433-446.
[3] “1. Introduction”, Page 2 of 15, Lines 68-70:
“SKOV3 has been an extensively used model for HGSOC, although recent evidence shows that this cell line represents phenotype of ovarian clear cell adenocarcinoma [14].”.
At that stage, the authors should note that the ovarian cancer-resistant cell line SKOV3 was recently utilized as a parental cell line for enriching stem cells using previously published culture methods. The findings indicated a correlation between BRCA1 expression and drug resistance genes in ovarian cancer tissues. Moreover, the data revealed a novel mechanism by which BRCA1 mediates autophagy-related regulation of cisplatin resistance. However, the specific pathway through which autophagy regulates and influences drug resistance remains unclear.
Recommended reference: You Y, et al. BRCA1 affects the resistance and stemness of SKOV3-derived ovarian cancer stem cells by regulating autophagy. Cancer Med. 2019;8(2):656-668.
Author Response
We sincerely appreciate the Reviewer insightful comments and recommendations, which have improved the quality of our work. Below, we addressed the Reviewer suggestions point by point:
1. We included the information about the cost of ovarian cancer treatment, as well as the proposed reference, in Lines 36–42:
“Ovarian cancer is one of the most lethal gynecological malignancies, it ranks third in incidence among gynecological cancers but first in mortality [1,2]. Lack of robust screening tests for early detection not only reduces the effectiveness of treatment, but also significantly increases its cost. This is one of the reasons why ovarian cancer ranks as the most expensive malignancy to treat among all types of cancer. In the United States, the economic toll of treatment per patient is substantial, with expenses ranging from approximately $80,000 in the first year of care to $100,000 in the terminal stages of the disease [3].”
2. Additional information about the importance of PI3K pathway in ovarian cancer was provided in the Results section rather than the Introduction section. This location ensures better relevance, as it directly relates to the mutations in the PI3K pathway genes observed in our cell line (Lines 153-156):
“Upregulation of the PI3K pathway is a common feature for EOC and leads to chemoresistance acquisition and cell cycle progression. Inhibition of this pathway induces genome instability and mitotic catastrophe by reducing the activity of Aurora kinase B [16].”
3. We expanded the description of the SKOV3 cell line by adding information about its ability to transform into stem-like cells Lines 76-79:
“SKOV3 is versatile, as it grows fast, is tolerant to environmental stresses, and its cultivation does not require especial growth conditions or expensive supplements. Furthermore, this cell line can be used as a parental cell line for stem-like cells isolation due to its high levels of stemness markers and chemoresistance [16,17].”
Reviewer 2 Report
Comments and Suggestions for Authors
Thank you for submitting this interesting manuscript reporting on establishing a new high grade serious cell ovarian cancer cell line which could help to further understand ovarian cancer and perform more experiments for clinical and biological use.
The manuscript was well written and text was clear and easy to read. The study design was appropriate to answer the research question and the conclusions were supported by the evidence presented. The methods were appropriate. The study would significantly add to our knowledge. However the results of the study should be interpreted cautiously due to some challenges and limitations which should be addressed in the discussion in details including the cell line strain have not been generated or validated in other independent laboratories. Future directions and recommendations on how to validate the biological and clinical characteristics of the cell line and its clinical utility and applications.
Author Response
We thank the Reviewer for the encouraging feedback. We appreciate the recognition of the potential impact of our study and the recommendation for pointing out its limitations. We added this information in the Discussion section in Lines 308-313:
“Our results show that OVAR79 cells exhibit an epithelial phenotype, including some features related to stemness as well as a more aggressive phenotype. Moreover, OVAR79 cells are more sensitive to cisplatin but more resistant to carboplatin compared with SKOV3 and OVCAR3 cells, as well as more resistant to paclitaxel compared to OVCAR3. However, for greater reliability, these results should be validated in other independent laboratories.”
Reviewer 3 Report
Comments and Suggestions for Authors
The strengths of the submitted study are the comprehensive description of the methodology and the satisfactory presentation of the results. The weaknesses are the limited number of cell lines used for comparison (only two) and the rudimentary discussion, which is restricted to approximately 20 lines.
It would be advisable to expand the comparison to include 3–4 additional ovarian cancer (OC) cell lines. If this is not feasible, the limited number of cell lines for comparison should be explicitly acknowledged as a limitation of the study.
Even if not obtained through original experiments, comparative characteristics of the novel cell line against established cell lines should be provided, for example, in a table.
Regarding the brief discussion section:
The landscape of ovarian cancer cell lines is not sufficiently described (see Domcke S, et al. Nat Commun. 2013. PMID: 23839242).
The consequences of different mutations for reconstructing an OC model ex vivo are only minimally addressed (see Nelson L, et al. Nat Commun. 2020. PMID: 32054838).
The potential pitfalls related to the source of the cell line (in this case, ascites instead of tumor) should be acknowledged (see Rickard BP, et al. Cancers. 2021. PMID: 34503128; Latifi A, et al. PLoS One. 2012. PMID: 23056490; Ding Y, et al. Mol Oncol. 2021. PMID: 34060699).
Minor issue: Ref. 25 is incompletely cited.
Author Response
We appreciate the Reviewer’s detailed and constructive feedback on our manuscript. Below we addressed the comments point by point:
Comment 1: “It would be advisable to expand the comparison to include 3–4 additional ovarian cancer (OC) cell lines. If this is not feasible, the limited number of cell lines for comparison should be explicitly acknowledged as a limitation of the study. Even if not obtained through original experiments, comparative characteristics of the novel cell line against established cell lines should be provided, for example, in a table.”
Response: We pointed out the limitation of our study and added the new chapter of the Results section, which includes a table of comparative characteristics of the novel cell line against the most popular ovarian cancer cell lines (Lines 242-274):
“2.7. Comparison of OVAR79 to other ovarian cancer cell lines
The majority of ovarian cancer research utilizes cell lines such as SKOV3, OVCAR3, A2780, and CaOV3 [41]. In our study, we compared the features of the OVAR79 cell line with SKOV3 and OVCAR3 cell lines through direct experiments. To address the limitation of using a restricted number of cell lines, we referred to published data to include comparative characteristics of A2780 and CaOV3 cell lines. We expanded the comparison by incorporating key aspects of cell line characteristics, including the patient diagnosis, origin of the cell line, mutational burden and chromosomal instability, proliferation and migration rates, and epithelial-mesenchymal transition status (Table 1). Despite the limited set of descriptive characteristics, all these cell lines are completely different and do not duplicate each other. At the same time, it is well known that some of the most commonly used ovarian cancer cell lines do not represent HGSOC, despite their frequent use as HGSOC models in numerous studies. In summary, this study highlights the importance of accurately characterizing new cell lines and carefully selecting a disease model for each research task based on the specific characteristics of the cell line.
Table 1. Comparison of OVAR79 cell line characteristics with the most commonly used ovarian cancer cell lines.
|
Feature |
OVAR79 |
SKOV3 |
OVCAR3 |
CaOV3 |
A2780 |
|
Origin |
Ascitic fluid |
Ascitic fluid [42] |
Ascitic fluid [43] |
Tumor [44] |
Tumor [45] |
|
Diagnosis |
HGSOC |
Clear cell carcinoma [15] |
HGSOC [46] |
Possibly HGSOC [41] |
Endometrioid ovarian carcinoma [47] |
|
TP53 mutation |
Wild-type |
Wild-type [41] or truncating mutation [47] |
Mutated [46] |
Mutated [41] |
Wild-type [41] |
|
PIK3CA mutations |
Mutated |
Mutated [47] |
Wild-type [48,49] |
Wild-type [47] |
Mutated [47] |
|
PTEN mutations |
Mutated |
Wild-type [47] |
Wild-type [48,49] |
Wild-type [47] |
Mutated [47] |
|
Proliferation rate |
High |
High |
Moderate |
High [50] |
High [51] |
|
Migration ability |
High |
High |
Low |
High [50] |
Low [52] |
|
Epithelial/Mesenchymal markers |
Epithelial with EMT potential |
Predominantly mesenchymal |
Epithelial [53] |
Epithelial [53] |
Epithelial [53] |
|
Chemoresistance |
Highly resistant to carboplatin, moderate sensitivity to cisplatin |
Resistant to paclitaxel, resistant to low doses of cisplatin |
Resistant to low doses of cisplatin |
Resistant to paclitaxel (comparable to SKOV3) [54] |
Sensitive to chemotherapy drugs of different classes [55] |
|
Stemness markers |
High NANOG, moderate CD44 |
Moderate |
High CD44 |
Moderate CD44, Bmi-1, low Dclk-1 [56] |
Low [57] |
|
Chromosomal instability |
High |
Moderate [58] |
High [58] |
Moderate [58] |
Low [58] |
”
Comment 2: “The landscape of ovarian cancer cell lines is not sufficiently described (see Domcke et al., 2013). The consequences of different mutations for reconstructing an OC model ex vivo are only minimally addressed (see Nelson et al., 2020). The potential pitfalls related to the source of the cell line (ascites instead of tumor) should be acknowledged (see Rickard et al., 2021; Latifi et al., 2012; Ding et al., 2021).”
Response: We improved the text of the Discussion section by adding more information about the landscape of ovarian cancer cell lines and the importance of gene mutations in each of them, and also described the main characteristics that distinguish tumor cells obtained from ascitic fluid compared with a solid tumor in Lines 276-299:
“Despite the frequent occurrence of the disease, cell lines representing HGSOC are often under-characterized and sometimes misidentified. The main genetic characteristics of HGSOC include TP53 mutations, homologous recombination deficiency, and a high level of genomic alterations [41,46]. HGSOC is also characterized by the expression of epithelial phenotype markers such as MUC16 (CA125), MSLN, PAX8, and KRT7 [49].
HGSOC is a markedly heterogeneous disease with significant diversity. Individual subtypes present exclusive features, but these may not always correspond to strict boundaries. For example, the presence or absence of a TP53 mutation alone is not a strict criterion for determining the subtype of a tumor. A biobank of ovarian cancer primary cell cultures, created by Nelson L. et al., revealed that genomic instability (the main characteristic of HGSOC) can be maintained even in the presence of wild-type TP53. Furthermore, it became evident that, despite having the same diagnosis, each tumor represents a unique combination of genetic alterations and mutations in a wide spectrum of genes [50]. It is worth noting that, despite their high genetic similarity, cancer cells from ascites differ from the original ovarian tumor cells. This difference arises because ascites has distinct physical properties and a unique cellular and molecular context compared to the solid tumor, creating selective conditions for clones capable of surviving without adhesion. Moreover, cancer cells in ascites can form spheroids — spherical cell conglomerates, that exhibit higher metastatic potential, increased expression of genes related to chemoresistance, and upregulation of the oxidative phosphorylation pathway [51–53]. Another important point is that cancer cells derived from ascites tend to display a more mesenchymal phenotype, which contributes to their higher metastatic potential, compared to the original tumor cells [54]. Therefore, the origin of a cancer cell line should be carefully taken into account.”
Comment 3: “Minor issue: Ref. 25 is incompletely cited.”
Response: Thank you for pointing out this oversight. We revised our reference list. After the first round of review, the reference [25] became [28] (Lines 519-521):
“Łukomska, A.; Menkiszak, J.; Gronwald, J.; Tomiczek-Szwiec, J.; Szwiec, M.; Jasiówka, M.; Blecharz, P.; Kluz, T.; Stawicka-Niełacna, M.; Mądry, R.; et al. Recurrent Mutations in BRCA1, BRCA2, RAD51C, PALB2 and CHEK2 in Polish Patients with Ovarian Cancer. Cancers (Basel) 2021, 13, doi:10.3390/cancers13040849.”
Round 2
Reviewer 2 Report
Comments and Suggestions for Authors
Thank you for submitting this revised and improved manuscript that sufficiently responded to my concerns.
Reviewer 3 Report
Comments and Suggestions for Authors
I appreciate the careful correction of the manuscript. The Table 1 is informative and easy to read. The discussion part improved greatly. Congratulations!